# Process-constrained batch Bayesian Optimisation

**Pratibha Vellanki**[1]**, Santu Rana**[1]**, Sunil Gupta**[1]**, David Rubin**[2]
**Alessandra Sutti**[2]**, Thomas Dorin**[2]**, Murray Height**[2]**,Paul Sandars**[3]**, Svetha Venkatesh**[1]

[1]Centre for Pattern Recognition and Data Analytics

Deakin University, Geelong, Australia

$\left[\texttt{pratibha.vellanki,santu.rana,sunil.gupta,svetha.venkatesh@deakin.edu.au}\right]$

[2]Institute for Frontier Materials, GTP Research

Deakin University, Geelong, Australia

$\left[\texttt{d.rubindecelisleal,alessandra.sutti,thomas.dorin,murray.height@deakin.edu.au}\right]$

[3]Materials Science and Engineering, Michigan Technological University, USA

$\left[\texttt{sanders@mtu.edu}\right]$

## Abstract

Prevailing batch Bayesian optimisation methods allow all control variables to be freely altered at each iteration. Real-world experiments, however, often have physical limitations making it time-consuming to alter all settings for each recommendation in a batch. This gives rise to a unique problem in BO: in a recommended batch, a set of variables that are expensive to experimentally change need to be fixed, while the remaining control variables can be varied. We formulate this as a *process-constrained batch Bayesian optimisation* problem. We propose two algorithms, *pc-BO(basic)* and *pc-BO(nested)*. *pc-BO(basic)* is simpler but lacks convergence guarantee. In contrast pc-BO(nested) is slightly more complex, but admits convergence analysis. We show that the regret of *pc-BO(nested)* is sublinear. We demonstrate the performance of both *pc-BO(basic)* and *pc-BO(nested)* by optimising benchmark test functions, tuning hyper-parameters of the SVM classifier, optimising the heat-treatment process for an Al-Sc alloy to achieve target hardness, and optimising the short polymer fibre production process.

## 1 Introduction

Experimental optimisation is used to design almost all products and processes, scientific and industrial, around us. Experimental optimisation involves optimising input control variables in order to achieve a target output. Design of experiments (DOE) [16] is the conventional laboratory and industrial standard methodology used to efficiently plan experiments. The method is rigid - not adaptive based on the completed experiments so far. This is where Bayesian optimisation offers an effective alternative.

Bayesian optimisation [13, 17] is a powerful probabilistic framework for efficient, global optimisation of expensive, black box functions. The field is undergoing a recent resurgence, spurred by new theory and problems and is impacting computer science broadly - tuning complex algorithms [3, 22, 18, 21], combinatorial optimisation [24, 12], reinforcement learning [4]. Usually, a prior belief in the form of Gaussian process is maintained over the possible set of objective functions and the posterior is the refined belief after updating the model with experimental data. The updated model is used to seek the most promising location of function extrema by using a variety of criteria, e.g. expected improvement (EI), and upper confidence bound (UCB). The maximiser of such a criteria function is then recommended for the function evaluation. Iteratively the model is updated and recommendations are made till the target outcome is achieved. When concurrent function evaluations are possible, Bayesian optimisation returns multiple suggestions, and this is termed as the *batch*

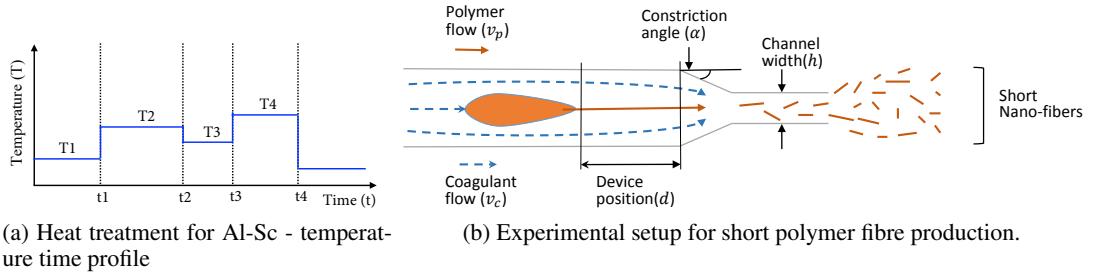

(a) Heat treatment for Al-Sc - temperature time profile

(b) Experimental setup for short polymer fibre production.

Figure 1: Examples of real-world applications requiring process constraints.

*setting.* Bayesian optimisation with batch setting has been investigated by [10, 5, 6, 9, 1] wherein different strategies are used to recommend multiple settings at each iteration. In all these methods, all the control variables are free to be altered at each iteration. However, in some situations needing to change all the variables for a single batch may not be efficient and this leads to the motivation of our *process-constrained Bayesian optimisation.*

This work has been directly influenced from the way experiments are conducted in many real-world scenarios with a typical limitation on resources. For example, in our work with metallurgists, we were given a task to find the optimal heat-treatment schedule of an alloy which maximises the strength. Heat-treatment involves taking the alloy through a series of exposures to different temperatures for a variable amount of durations as shown in Figure 1a. Typically, a heat treatment schedule can last for multiple days, so doing one experiment at a time is not efficient. Fortunately, a furnace is big enough to hold multiple samples at the same time. If we have to perform multiple experiments in one batch yet using only one furnace, then we must design our Bayesian optimisation recommendations in such a way that the temperatures across a batch remain the same, whilst still allowing the durations to vary. Samples would be put in the same oven, but would be taken out after different elapsed time for each step of the heat treatment. Similar examples abound in other domains of process and product design. For short polymer fibre production a polymer is injected axially within another flow of a solvent in a particular geometric manifold [20]. A representation of the experimental setup marked with the parameters involved is shown in Figure 1b. When optimising for the yield it is generally easy to change the flow parameters (pump speed setting) than changing the device geometry (opening up the enclosure and modifying the physical configuration). Hence in this case as well, it is beneficial to recommend a batch of suggested experiments at a fixed geometry but allowing flow parameters to vary. Many such examples where the batch recommendations are constrained by the processes involved have been encountered by the authors in realising the potential of Bayesian optimisation for real-world applications.

To construct a more familiar application we use the hyper-parameter tuning problem for Support Vector Machines (SVM). When we use parallel tuning using batch Bayesian optimisation, it may be useful if all the parallel training runs finished at the same time. This would require fixing the cost parameter, while allowing the the other hyper-parameters to vary. Whist this may or may not be a real concern depending on the use cases, we use it here as a case study.

We formulate this unique problem as *process-constrained batch Bayesian optimisation*. The recommendation schedule needs to constrain a set of variables corresponding to control variables that are experimentally expensive (time, cost, difficulty) to change (*constrained set*) and varies all the remaining control variables *(unconstrained set)*. Our approach involves incorporating constraints on stipulated control parameters and allowing the others to change in an unconstrained manner. The mathematical formulation of our optimisation problem is as follows.

$$x^* = \text{argmax}_{x \in \mathcal{X}} f(x)$$

and we want a batch Bayesian optimisation sequence

$$\{\{x_{t,0}, x_{t,1}, ..., x_{t,K-1}\}\}_{t=1}^{T} \text{ such that } \forall t \text{ and } x_{t,k} = [x_{t,k}^{uc} x_{t,k}^{c}],$$

$$x_{t,k}^{c} = x_{t,k'}^{c}, \forall k, \ k^{'} \in [0, ..., K-1]$$

Where $x_{t,k}^{c}$ is the $k^{\text{th}}$ constrained variable in $t^{\text{th}}$ batch and similarly $x_{t,k}^{uc}$ is the $k^{\text{th}}$ unconstrained variable in the $t^{\text{th}}$ batch. $T$ is the total number of iterations and $K$ is the batch-size.

We propose two approaches to the solve this problem: *basic process-constrained Bayesian optimisation (pc-BO(basic))* and *nested process-constrained batch Bayesian optimisation (pc-BO(nested))*. *pc-BO(basic)* is an intuitive modification motivated by the work of [5] and *pc-BO(nested)* is based on a nested Bayesian optimisation method we will describe in section 3. We formulate the algorithms *pc-BO(basic)* and *pc-BO(nested)*, and for *pc-BO(nested)* we present the theoretic analysis to show that the average regret vanishes superlinearly with iterations. We demonstrate the performance of *pc-BO(basic)* and *pc-BO(nested)* on both benchmark test functions and real world problems that involve hyper-parameter tuning for SVM classification for two datasets: breast cancer and biodegradable waste, the industrial problem of heat treatment process for an Aluminium-Scandium (Al-Sc) alloy, and another industrial problem of short polymer fibre production process.

## 2 Related background

### 2.1 Bayesian optimisation

Bayesian optimisation is a sequential method of global optimisation of an expensive and unknown black-box function $f$ whose domain is $\mathcal{X}$, to find its maxima $x^* = \operatorname*{argmax}_{x \in \mathcal{X}} f(x)$ (or minima). It is especially powerful when the function is expensive to evaluate and it does not have a closed-form expression, but it is possible to generate noisy observations from experiments.

The Gaussian process (GP) is commonly used as a flexible way to place a prior over the unknown function [14]. It is are completely described by the mean function $m(x)$ and the covariance function $k(x, x')$ and they imply our belief and uncertainties about the objective function. Noisy observations from the experiments are sequentially appended into the model, that in turn updates our belief about the objective function.

The acquisition function is a surrogate utility function that takes a known tractable closed form and allows us to choose the next query point. It is maximised in the place of the unknown objective function and constructed such that it balances between exploring regions of high value (mean) and exploiting regions of high uncertainties (variances) across the objective function.

Gaussian process based Upper Confidence Bound (GP-UCB) proposed by [19] is one of the acquisition functions which is shown to achieve sublinear growth in cumulative regret. It is define at $t^{\text{th}}$ iteration as

$$\alpha_{GP-UCB}^t(x) = \mu_{t-1}(x) + \sqrt{\beta_t}\sigma_{t-1}(x) \tag{1}$$

where, $v = 1$ and $\beta_t = 2log(t^{d/2+2}\pi^2/3\delta)$ is the confidence parameter, wherein $t$ denotes the iteration number, $d$ represents the dimensionality of the data and $\delta \in (0, 1)$. We are motivated by GP-UCB based methods. Although our approach can be intuitively extended to other acquisition function, we do not explore this in the current work.

### 2.2 Batch Bayesian optimisation methods

The GP exhibits an interesting characteristic that its predictive variance is dependent on only the input attributes while updating its mean requires knowledge about the outcome of the experiment. This leads us to a direction of strategies for multiple recommendations. There are several batch Bayesian optimisation algorithms for an unconstrained case. GP-BUCB by [6] recommends multiple batch points using the UCB strategy and the aforementioned characteristic. To fill up a batch, it updates the variances with the available attribute information and appends the outcomes temporarily by substituting them with most recently computed posterior mean. A similar strategy is used in the GP-UCB-PE by [5] that optimises the unknown function by incorporating some batch elements where uncertainty is high. GP-UCB-PE computes the first batch element by using the UCB strategy and recommends the rest of the points by relying on only the predictive variance, and not the mean. It has been shown that for these GP-UCB based algorithms the regret can be bounded tighter than the single recommendation methods. To the best of our knowledge these existing batch Bayesian optimisation techniques do not address the process-constrained problem presented in this work. The algorithms proposed in this paper are inspired by the previous approaches but address it in context of a process-constrained setting.

### 2.3 Constrained-batch vs. constrained-space optimisation

We refer to the parameters that are not allowed to change (eg. temperatures for heat treatment, or device geometry for fibre production) as constrained set and the other parameters (heat treatment durations or flow parameters) as unconstrained set. We emphasise that our usage of *constraint* differs from the problem settings presented in literature, for example in [2, 11, 7, 8], where the parameters values are constrained or the function evaluations are constrained by inequalities. In the problem setting that we present, all the parameters exist in unconstrained space; for each individual batch, the constrained variables should have the same value.

## 3 Proposed method

We recall the maximisation problem from Section 1 as $x^* = \mathrm{argmax}_{x \in \mathcal{X}} f(x)$. In our case $\mathcal{X} = \mathcal{X}^{uc} \cup \mathcal{X}^c$, where $\mathcal{X}^c$ is the constrained subspace and $\mathcal{X}^{uc}$ is the unconstrained subspace.

---

**Algorithm 1** pc-BO(basic): Basic process-constrained pure exploration batch Bayesian optimisation algorithm.

---

**while** $(t < MaxIter)$

   $x_{t,0} = \left[x_{t,0}^{uc} x_{t,0}^c\right] = \mathrm{argmax}_{x \in \mathcal{X}} \alpha^{GP-UCB}\left(x_{t,0} \mid D\right)$

   **for** $k = 1, .., K-1$

      $x_{t,k}^{uc} = \mathrm{argmax}_{x^{uc} \in \mathcal{X}^{uc}} \sigma\left(x_{t,k}^{uc} \mid D, x_{t,0}^c, \left\{x_{t,k'}^{uc}\right\}^{k'<k}\right)$

   **end**

   $D = D \cup \left\{\left[x_{t,k}^{uc} x_{t,1}^c\right], f\left(\left[x_{t,k}^{uc} x_{t,1}^c\right]\right)\right\}_{k=0}^{K-1}$

**end**

---

**Algorithm 2** pc-BO(nested): Nested process-constrained batch Bayesian optimisation algorithm.

---

**while** $(t < MaxIter)$

   $x_t^c = \mathrm{argmax}_{x^c \in \mathcal{X}^c} \alpha_c^{GP-UCB}\left(x_t^c \mid D_O\right)$

   $x_{t,0}^{uc} = \mathrm{argmax}_{x^{uc} \in \mathcal{X}^{uc}} \alpha_{uc}^{GP-UCB}\left(x_t^{uc} \mid D_I, x_t^c\right)$

   **for** $k = 1, ..., K\text{-}1$

      $x_{t,k}^{uc} = \mathrm{argmax}_{x^{uc} \in \mathcal{X}^{uc}} \sigma_{uc}\left(x_t^{uc} \mid D_I, x_t^c, \left\{x_{t,k'}^{uc}\right\}^{k'<k}\right)$

   **end**

   $D_O = D_O \cup \left\{x_t^c, f\left(\left[(x_t^{uc})^+ x_t^c\right]\right)\right\}$

   $D_I = D_I \cup \left\{\left[x_{t,k}^{uc} x_t^c\right], f\left(\left[x_{t,k}^{uc} x_t^c\right]\right)\right\}_{k=0}^{K-1}$

**end**

---

A naïve approach to solving the process is to employ any standard batch Bayesian optimisation algorithm where the first member is generated and then subsequent members are filled up by setting the constraint variables to that of the first member. We describe this approach as the *basic process-constrained pure exploration batch Bayesian optimisation (pc-BO(basic))* algorithm as detailed in algorithm 1, where $\alpha^{GP-UCB}(x \mid D)$ is the acquisition function as defined in Equation 1. We note that *pc-BO(basic)* is an improvisation over the work of [5]. During each iteration, the first batch element is recommended using the UCB strategy. The remaining batch elements, as in GP-UCB-PE, are generated by updating the posterior variance of the GP, after the constrained set attributes are fixed to those of the first batch element.

We provide an alternate formulation via a nested optimisation problem called *nested process-constrained batch Bayesian optimisation (pc-BO(nested))* with two stages. For each batch, in the outer stage optimisation is performed to find the optimal values of the constrained variables and in the inner stage optimisation is performed to find optimal values of the unconstrained variables. The algorithm is detailed in algorithm 2, where $\alpha_c^{GP-UCB}(x \mid D)$ is the acquisition function for the outer stage, and $\alpha_{uc}^{GP-UCB}(x \mid D)$ is the acquisition function for the inner stage as defined in Equation 1, and $(x_t^{uc})^+ = \mathrm{argmax}_{x_t^{uc} \in \left\{x_{t,k}^{uc}\right\}_{k=0}^{K-1}} f\left([x_t^{uc} x_t^c]\right)$, is the unconstrained batch parameter that yields the best target goal for the given constrained parameter $x^c$. We are able to analyse the convergence of

*pc-BO(nested).* It can be expected that in some cases the performance of the pc-BO(basic) and pc-BO(nested) are close. The pc-BO(basic) method maybe considered simpler, but it lacks guaranteed convergence.

## 3.1 Convergence analysis for pc-BO(nested)

We now present the analysis of the convergence of pc-BO(nested) as described in Algorithm 2. The outer stage optimisation problem for $x^c$ and observation $D_o$ is expressed as follows.

$$
\begin{aligned}
(x^c)^* &= \text{argmax}_{x^c \in \mathcal{X}^c} g(x^c), \\
\text{where,} \quad g(x^c) &\triangleq \max_{x^{uc} \in \mathcal{X}^{uc}} f\left([x^{uc} x^c]\right) \\
&\simeq \max_{x^{uc} \in X^{uc}} f([x^{uc} x^c]) = f([(x^{uc})^+ x^c]), \\
\text{where,} \quad X^{uc} &\triangleq \{\{x_{t,0}, x_{t,1}, ..., x_{t,K-1}\}\}_{t=1}^T \quad \text{such that,} \ x_{t,k}^c = x^c, \\
D_O &\triangleq \left\{ x_t^c, \ f\left(\left[(x_{t,k}^{uc})^+ x_{t,}^c\right]\right) \right\}_{t=1}^T
\end{aligned}
$$

And the inner stage optimisation problem for $x^{uc}$ and observation $D_I$ is expressed as follows.

$$
\begin{aligned}
(x^{uc})^* &= \text{argmax}_{x^{uc} \in \mathcal{X}^{uc}} h\left(x^{uc}\right), \\
\text{where,} \quad h(x^{uc}) &\triangleq f\left([x^{uc} x^c]\right) \\
D_I &\triangleq \left\{ \left\{ \left[x_{t,k}^{uc} x_t^c\right], f\left(\left[x_{t,k}^{uc} x_t^c\right]\right) \right\}_{k=0}^{K-1} \right\}_{t=1}^T
\end{aligned}
$$

This is solved using a Bayesian optimisation routine. Here, $(x^{uc})^+$ is the unconstrained batch parameter that yields the best target goal for the given constrained parameter $x^c$. Unfortunately as $g(x^c)$ is not easily measurable, we use $f([(x^{uc})^+ x^c])$ as an approximation to it. To address this we use a provable batch Bayesian optimisation such as GP-UCB-PE [5] in the inner stage. The loops are performed together where in each iteration $t$, the outer loop first recommends a single recommendation of $x_t^c$ and then the inner loop suggests a batch, $\{x_{t,k}^{uc}\}_{k=1}^K$. Combining them we get process-constrained set of recommendations. We show that together these two Bayesian optimisation loops converge to the optimal solution.

Let us denote $(x_t^{uc})^+ = \text{argmax}_{\boldsymbol{x}^{uc} \in \{\boldsymbol{x}_k^{uc}\}_{k=1}^K} f([x^{uc} x_t^c])$. Following that we can write $g(x^c)$ as,

$$
\begin{aligned}
g(x^c) &= f\left(\left[(x_t^{uc})^* x_{t,}^c\right]\right) = f\left(\left[(x_t^{uc})^+ x_{t,}^c\right]\right) + f\left(\left[(x_t^{uc})^* x_{t,}^c\right]\right) - f\left(\left[(x_t^{uc})^+ x_{t,}^c\right]\right) \\
&= f\left(\left[(x_t^{uc})^+ x_{t,}^c\right]\right) + r_t^{uc}
\end{aligned}
\tag{2}
$$

where $r_t^{uc}$ is the regret of the inner loop.

The observational model is given as

$$
y^c = g(x^c) + \epsilon = f\left(\left[(x_t^{uc})^+ x_{t,}^c\right]\right) + r_t^{uc} + \epsilon \qquad \text{where } \epsilon \sim N(0, \sigma^2)
\tag{3}
$$

**Lemma 1.** *For regret of the inner loop,* $\sum_{t=1}^T \left(r_t^K\right)^2 \leq \beta_1^{uc} C_1^{uc} \gamma_T^{uc} + \frac{\pi^2}{6}$

*Proof.* As we use GP-UCB-PE for unconstrained parameter optimisation, we can say that the regret $r_t^K = \min r_t^k \quad \forall k = 0, ..., K-1$ (Lemma 1, [5]). Hence, $r_t^K \leq r_t^0 \leq 2\sqrt{\beta_1}\sigma_t^0$. Now, even though every batch recommendation for $x^c$ will always be run for one iteration only, the $\sigma_t^0(x_t)$ is computed from the updated GP. Hence the sum of $(\sigma_t^0)^2$ can be upper bounded by $\gamma_T$. Thus,

$$
\sum_{t=1}^T \left(r_t^K\right)^2 \leq \beta_1^{uc} C_1^{uc} \gamma_T^{uc} + \frac{\pi^2}{6}
\tag{4}
$$

Here, $\beta_1 = 2log(1^{d/2+2}\pi^2/3\delta)$ is the confidence parameter; $C_1 = 8/log(1 + \sigma^{-2})$; $\gamma_T = \max_{A \in \mathcal{X}^c, |A|=T} I(y_A : f_A)$ assuming $y = f + \epsilon$, where $\epsilon \sim \mathcal{N}(0, \sigma^2/2)$ is the maximum information gain after $T$ rounds. (Please see supplementary material 5 for derivation) ☐

**Lemma 2.** *For the variance of* $r_t^{uc}$ *has the order of* $\sigma_{r_t}^2 \sim \mathcal{O}(C_1^{uc}\beta_1^{uc}\gamma_t^{uc} + C_2^{uc})$

*Proof.* We use PE algorithm [5] to compute *K*-recommendation, hence the variance of the regret $r_t^{uc}$ can be bounded above by

$$\sigma_{r_t^{uc}}^2 \leq \mathbb{E}((r_t^{uc})^2) \leq \mathbb{E}\left(\frac{1}{t}\sum_{t'=0}^{t}(r_{t'}^{uc})^2\right) = \mathbb{E}\left(\frac{1}{t}\sum_{t'=0}^{t}\min_{k<K}(r_{t'k}^{uc})^2\right)$$

The second inequality holds since on an average the gap $r_t^{uc} = g(x^c) - f([(\boldsymbol{x}_t^{uc})^+\boldsymbol{x}^c])$ decreases with iteration $t$, $\forall \boldsymbol{x}^c \in \mathcal{X}^c$. From equation 3, equation 4 and using the Lemma 4 and 5 of [5] we can write

$$\mathbb{E}\left(\frac{1}{t}\sum_{t'=0}^{t}\min_{k<K}(r_{t'k}^{uc})^2\right) \sim \mathcal{O}(\frac{1}{t}C_1^{uc}\beta_1^{uc}\gamma_t^{uc} + C_2^{uc}) \tag{5}$$

for some $C_1^{uc}, C_2^{uc} \in \mathbb{R}$. $\gamma_t$ is the maximum information gain over $t$ samples. This concludes the proof. $\qquad\square$

The following lemma guarantees an existence of a finite $T_0$ after which the noise variance coming from the inner optimisation loop becomes smaller than the noise in the observation model.

**Lemma 3.** $\exists T_0 < \infty$ *for which* $\sigma_{r_{T_0}^{uc}}^2 \leq \sigma^2$.

*Proof.* In Lemma 1,$C_1^{uc}, C_2^{uc}$ and $\beta_1^{uc}$ are fixed constant and $\gamma_{tK}^{uc}$ is sublinear in $t$. Therefore, any quantity of the form $M_1 \times \frac{1}{t}C_1^{uc}\beta_1^{uc}\gamma_t^{uc} + C_2^{uc}$ also decreases sublinearly with $t$ for $\forall M_1 \in \mathbb{R}$. Hence the lemma is proved. $\qquad\square$

Let us denote the instantaneous regret for the outer Bayesian optimisation loop as $r_t^c = g((x^c)^*) - g(x_t^c)$, we can write the average regret after $T$ iterations as,

$$\begin{aligned}
\bar{R}_T &= \frac{1}{T}\sum_{t=0}^{T}r_t^c \leq \frac{1}{T}\sum(2\sqrt{\beta_t^c}\sigma_{t-1}^c(\boldsymbol{x}_t^c) + \frac{1}{t^2}) \\
&\leq 2\sqrt{\frac{\beta_T^c \sum(\sigma_{t-1}^c(\boldsymbol{x}_t^c))^2}{T}} + \frac{1}{T}\sum\frac{1}{t^2}
\end{aligned} \tag{6}$$

using the Lemma 5.8 of [19] and Cauchy-Schwartz inequality.

**Lemma 4.** *For the outer Bayesian optimisation* $\lim_{T\to\infty}\bar{R}_T \to 0$

*Proof.* From the equation 6

$$\begin{aligned}
\bar{R}_T &\leq 2\sqrt{\frac{\beta_T^c \sum_{t=1}^{T}(\sigma_{t-1}^c(\boldsymbol{x}_t^c))^2}{T}} + \frac{1}{T}\sum_{t=1}^{T}\frac{1}{t^2} \\
&= 2\sqrt{\frac{\beta_T^c}{T}\left(\sum_{t=1}^{T_0}((\sigma_{t-1}^c(\boldsymbol{x}_t^c))^2 + \sum_{t=T_0+1}^{T}(\sigma_{t-1}^c(\boldsymbol{x}_t^c))^2\right)} + \frac{1}{T}\sum_{t=1}^{T}\frac{1}{t^2} \\
&\leq 2\sqrt{\frac{\beta_T^c}{T}(A_{T_0} + B_T)} + \frac{1}{T}\sum_{t=1}^{T}\frac{1}{t^2}
\end{aligned} \tag{7}$$

We then show that $A_{T_0}$ is upper bounded by a constant irrespective of $T$ as long as $T \geq T_0$ and $B_T$ is sublinear with $T$. $\beta_T^c$ is sublinear in $T$ and $\lim_{T\to\infty}\sum_{t=1}^{T}\frac{1}{t^2} = \frac{\pi^2}{6}$. Hence the right hand side vanishes as $T \to \infty$. The details of the proof is presented in the supplementary material. $\qquad\square$

However, in reality using regret as the upper bound on $r_t^{uc}$ is not necessary, as a tighter upper bound may exist when we know the maximum value of the function[1] and we can safely alter the upper bound as,

$$r_t^{uc} \leq min(f^{max} - f([(\boldsymbol{x}_t^{uc})^+\boldsymbol{x}^c]), 2\sqrt{\beta_1}\sigma_{t-1}^{uc}(\boldsymbol{x}_0^{uc})) \tag{8}$$

The above results holds since Lemma 2 still holds.

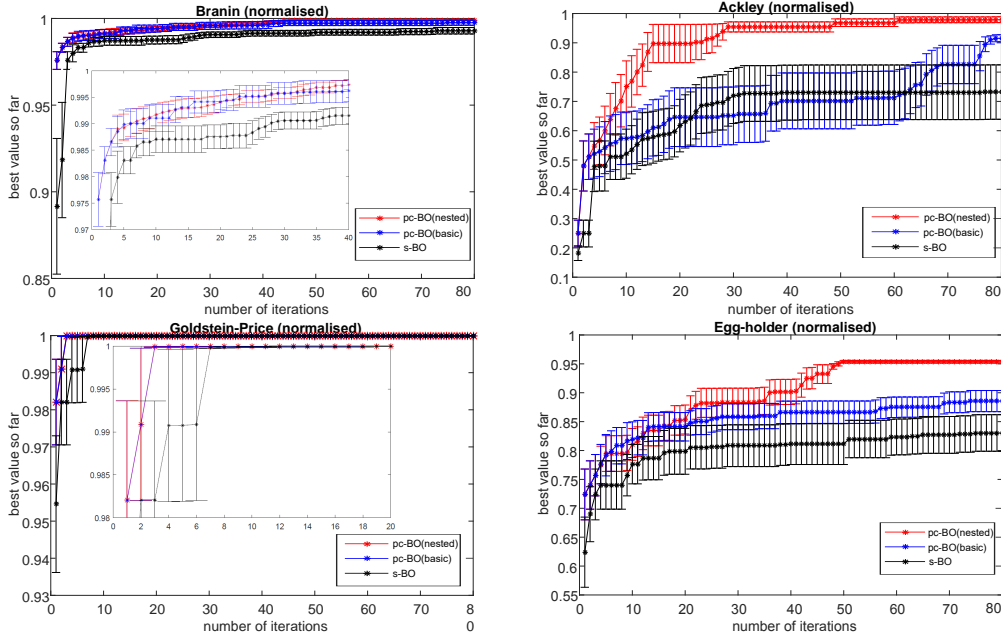

Figure 2: Synthetic test function optimisation using pc-BO(nested), pc-BO(basic) and s-BO. The zoomed area on the respective scale is shown for Branin and Goldstein-Price.

## 4 Experiments

We conducted a set of experiments using both synthetic data and real data to demonstrate the performance of pc-BO(basic) and pc-BO(nested). To the best of our knowledge, there are no other methods that can selectively constrain parameters in each batch during Bayesian optimisation. Further, we also show the results for the test function optimisation using *sequential* BO (s-BO) using GP-UCB.

The code is implemented in MATLAB and all the experiments are run on an Intel CPU E5-2640 v3 @2.60GHz machine. We use the squared exponential distance kernel. To show the performance, we plot the results as the best outcome so far against the number of iterations performed. The uncertainty bars in the figures pertain to 10 runs of BO algorithms with different initialisations for a batch of 3 recommendations. The errors bars show the standard error and the graph shows the mean *best outcome* until the respective iteration.

### 4.1 Benchmark test function optimisation

In this section, we use benchmark test functions and demonstrate the performance of pc-BO(basic) and pc-BO(nested). We apply the test functions by constraining the second parameter and finding the best configuration across the first parameter (unconstrained). The Branin, Ackley, Goldstein-Price and the Egg-holder functions were optimised using pc-BO(basic) and pc-BO(nested), and the results are shown in Figure 2. From the results, we note that the pc-BO(nested) is marginally better or similar in performance when compared with pc-BO(basic). It also shows that batch Bayesian optimisation is more efficient in terms of number of iterations than a purely sequential approach for the problem at hand.

### 4.2 Hyper-parameter tuning for SVM

Support vector machines with RBF kernel require hyper-parameter tuning for Cost ($C$) and Gamma ($\gamma$). Out of these parameters, the cost is a critical parameter that trades off error for generalisation. Consider tuning SVM's in parallel. The cost parameter strongly affects the time required for training SVM. It would be inconvenient if one training process took much longer than the other. Thus constraining the cost parameter for a single batch maybe a good idea. We use our algorithms to tune

both the hyper-parameters $C$ and $\gamma$, at each batch only varying $\gamma$, but not $C$. This is demonstrated on the classification using SVM problem using two datasets downloaded from UCI machine learning repository: Breast cancer dataset (BCW) and Bio-degradation dataset (QSAR).

BCW has 683 instances with 9 attributes each of the data, where the instances are labelled as benign or malign tumour as per the diagnosis. The QSAR dataset categorises 1055 chemicals with 42 attributes as ready or not ready biodegradable waste. The results are plotted as best accuracy obtained across number of iterations. We observe from the results in Figure 3, that pc-BO(nested) again performs marginally better than pc-BO(basic) for the BCW dataset. For the QSAR dataset, pc-BO(nested) higher accuracy with lesser iterations than what pc-BO(basic) requires.

### 4.3 Heat treatment for an Al-Sc alloy

Alloy casting involves heat treatment process - exposing the cast to different temperatures for select times, that ensures target hardness of the alloy. This process is repeated in steps. The underlying physics of heat-treatment of an alloy is based on nucleation and growth. During the nucleation process, "new phases" or precipitates are formed when clusters of atom self organise. This is a difficult stochastic process that happens at lower temperatures. These precipitates then diffuse together to achieve the requisite target alloy characteristics in the growth step. KWN [15, 23] is the industrial standard precipitation model for the kinetics of nucleation and growth steps. As a preliminary study we use this simulator to demonstrate the strength of our algorithm.

As explained in the introduction, it is cost efficient to test heat treatment in the real world by varying the time and keeping the temperature constrained in each batch. This will allow us to test multiple samples at one go in a single oven. We use the same constrains for our simulator driven study. We consider a two stage heat treatment process. The input to first stage is the alloy composition, the temperature and time. The nucleation output of this stage is input to the the second stage along with the temperature and time for the second stage. The final output is hardness of the material (strength in kPa). To optimise this two stage heat treatment process our inputs are $[T_1, T_2, t_1, t_2]$, where $[T_1, T_2]$ represent temperatures in *Celsius*, $[t_1, t_2]$ represent the time in *minutes* for each stage. Figure 4 shows the results of the heat-treatment process optimisation.

### 4.4 Short polymer fibre polymer production

Short polymer fibre production is a set of experiments we conducted in collaboration with material scientists at Deakin University. For production of short polymer fibres, a polymer rich fluid is injected coaxially into the flow of another solvent in a particular geometric manifold. The parameters included in this experiment are device position in *mm*, constriction angle in *degrees*, channel width in *mm*, polymer flow in *ml/hr*, and coagulant speed in *cm/s*. The final output, the combined utility is the distance of the length and diameter of the polymer from target polymer. The goal is to optimise the input parameters to obtain a polymer fibre of a desired length and diameter. As explained in the introduction, it is efficient to test multiple combinations of polymer flow and coagulant speed for a fixed geometric setup than in a single batch.

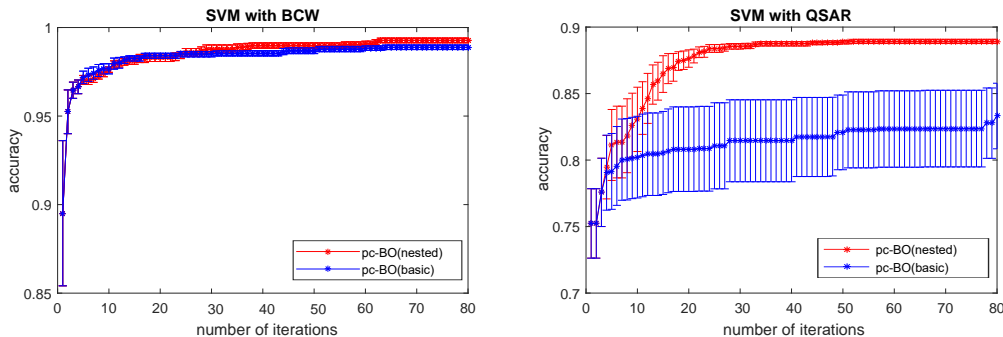

Figure 3: Hyper-parameter tuning for SVM based classification on Breast Cancer Data (BCW) and bio-degradable waste data (QSAR) using pc-BO(nested) and pc-BO(basic)

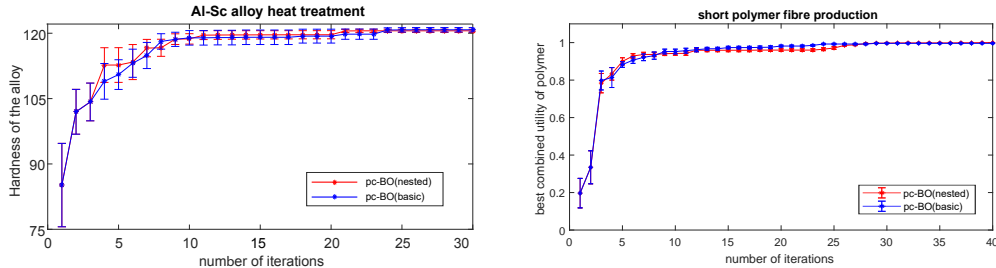

Figure 4: Results for heat-treatment and short polymer fibre production processes. (a) Experimental result for Al-Sc heat treatment profile for a two stage heat-treatment process using pc-BO(nested) and pc-BO(basic). (b) Optimisation for short polymer fibre production with position, constriction angle and channel width constrained for each batch. Polymer flow and coagulant speed are unconstrained. The optimisation is shown for pc-BO(nested) and pc-BO(basic) algorithms.

The parameters in this experiments are discrete, where every parameter takes 3 discrete values, except the constriction angle which takes 2 discrete values. Coagulant speed and polymer flow are unconstrained parameters and channel width, constriction angle and position are the constrained parameters. We conducted the experiment in batches of 3. The Figure 4 shows the optimisation results for this experiment over 53 iterations.

## 5 Conclusion

We have identified a new problem in batch Bayesian optimisation, motivated from physical limitations in real world experiments while conducting batch experiments. It is not feasible and resource-friendly to change all available settings in scientific and industrial experiments for a batch. We propose *process-constrained batch Bayesian optimisation* for such applications, where it is preferable to fix the values of some variables in a batch. We propose two approaches to solve the problem of process-constrained batches pc-BO(basic) and pc-BO(nested). We present analytical proof for convergence of pc-BO(nested). Synthetic functions, and real world experiments: hyper-parameter tuning for SVM, alloy heat treatment process, and short polymer fiber production process were optimised using the proposed algorithms. We found that pc-BO(nested) in each of these scenarios is either more efficient or equally well performing compared with pc-BO(basic).

## Acknowledgements

This research was partially funded by the Australian Government through the Australian Research Council (ARC) and the Telstra-Deakin Centre of Excellence in Big Data and Machine Learning. Prof Venkatesh is the recipient of an ARC Australian Laureate Fellowship (FL170100006).

## Footnotes

[1]e.g. for hyper-parameter tuning we know that maximum value of accuracy is 1.

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
