[Supplementary Material]

## Supplementary material

**Derivation for Lemma 1, 4**   As , $r_t^K \le r_t^0 \le 2\sqrt{\beta_1}\sigma_t^0$, we can say that

$$\sum_{t=1}^{T}\left(r_t^K\right)^2 \le \sum 4\beta_1^{uc}\left(\sigma_t^0\right)^2 + \sum_{t=1}^{T}\frac{1}{t^2}$$

$$\le 4\beta_1^{uc}\frac{c_1^{uc}\gamma_T^{uc}}{4} + \frac{\pi^2}{6} \qquad \text{(because } \sum_{t=1}^{T}\frac{1}{t^2} \le \frac{\pi^2}{6}\text{)}$$

$$= \beta_1^{uc}c_1^{uc}\gamma_T^{uc} + \frac{\pi^2}{6}$$

**Derivation for $A_{T_0}$ and $B_T$ in Lemma 3**   From Equation 7

$$\bar{R}_T \le 2\sqrt{\frac{\beta_T^c}{T}(A_{T_0}+B_T)} + \frac{1}{T}\sum_{t=1}^{T}\frac{1}{t^2} \tag{9}$$

For the first term,

$$A_{T_0} = \sum_{t=1}^{T_0}((\sigma_{t-1}^c(\boldsymbol{x}_t^c))^2$$

$$= \sum \sigma^2\overbrace{\left(\sigma_{r_{T_0}}^{-2}\,(\sigma_{t-1}^c\,(x_t^c))^2\right)}^{s^2} \tag{10}$$

Now, since $\sigma_{t-1}^c \le 1$,

$$s^2 = \sigma_{r_{T_0}}^{-2}\,(\sigma_{t-1}^c\,(x_t^c))^2$$

$$\le \sigma_{r_{T_0}}^{-2}$$

We know if $C_2 = \dfrac{\sigma_{r_{T_0}}^{-2}}{log\left(1+\sigma_{r_{T_0}}^{-2}\right)}$, then $s^2 \le C_2 log\left((1+s^2)\right)$

Thus we have from equation 10,

$$A_{T_0} = \sum \sigma^2 s^2$$

$$\le \sum \sigma^2 C_2 log\left(1 + \sigma_{r_{T_0}}^{-2}\,(\sigma_{t-1}^c)^2\right)$$

$$= \sum \sigma^2 \frac{\sigma_{r_{T_0}}^{-2}}{log\left(1+\sigma_{r_{T_0}}^{-2}\right)}log\left(1 + \sigma_{r_{T_0}}^{-2}\,(\sigma_{t-1}^c)^2\right)$$

$$= \sum_{t=1}^{T_0}\left(\frac{log\left(1+\sigma_{r_{T_0}}^{-2}\,(\sigma_{t-1}^c(\boldsymbol{x}_t^c))^2\right)}{log\left(1+\sigma_{r_{T_0}}^{-2}\right)}\right)$$

Since $\sigma_{t-1}^c(\boldsymbol{x}_t^c) \le 1$, we have $A_{T_0} \le \sum_{t=1}^{T_0}\left(\frac{log\left(1+\sigma_{r_{T_0}}^{-2}\times 1\right)}{log\left(1+\sigma_{r_{T_0}}^{-2}\right)}\right) = T_0$, which is a constant. Hence $\lim_{T\to\infty}\frac{\beta_T^c}{T}A_{T_0} \to 0$.

For the second term,

$$B_T = \sum_{T_0+1}^{T}((\sigma_{t-1}^c(\boldsymbol{x}_t^c))^2$$

$$\le \sum_{t=T_0+1}^{T}\left(\frac{log\left(1 + (\sigma_{r_{T_0}}^{-2} + \sigma^{-2})(\sigma_{t-1}^c(\boldsymbol{x}_t^c))^2\right)}{log\left(1 + (\sigma_{r_{T_0}}^{-2} + \sigma^{-2})\right)}\right)$$

From Lemma 2 after $T > T_0$ we know $\sigma_{r_{T_0}} < \sigma$, we have

$$
\begin{aligned}
B_T &\leq \sum_{t=T_0+1}^{T} \left( \frac{log\left(1 + (\sigma^{-2} + \sigma^{-2})(\sigma_{t-1}^c(\boldsymbol{x}_t^c))^2\right)}{log\left(1 + (\sigma^{-2} + \sigma^{-2})\right)} \right) \\
&= \sum_{t=T_0+1}^{T} \left( \frac{log\left(1 + 2\sigma^{-2}(\sigma_{t-1}^c(\boldsymbol{x}_t^c))^2\right)}{log\left(1 + 2\sigma^{-2}\right)} \right) \\
&= \sum_{t=T_0+1}^{T} \left( \frac{log\left(1 + \tilde{\sigma}^2(\sigma_{t-1}^c(\boldsymbol{x}_t^c))^2\right)}{log\left(1 + 2\sigma^{-2}\right)} \right) \\
&\leq \frac{1}{log\left(1 + 2\sigma^{-2}\right)} \tilde{\gamma}_T
\end{aligned}
$$

where $\tilde{\gamma}_T = \max_{A \in \mathcal{X}^c, |A|=T} I(y_A : f_A)$ assuming $y = f + \tilde{\epsilon}$, where $\tilde{\epsilon} \sim \mathcal{N}(0, \sigma^2/2)$. Since $\tilde{\gamma}_T$ grows sublinearly with $T$, especially for SE kernel $\tilde{\gamma}_T \sim \mathcal{O}((logT)^{d+1})$, it is easy to show that $\lim_{T \to \infty} \frac{\beta_T^c}{T} B_T \to 0$. Hence $\lim_{T \to \infty} \bar{R}_T \to 0$.