[Reviews · NeurIPS 2017]

Reviewer 1



The paper proposes a batch Bayesia Optimization algorithm for problems in which the queried parameters in a batch need to be the same. The authors first proposed a straight forward algorithm where the constrained parameters are chosen according to the first element in a batch. Then, the authors propose an nested algorithm where the outer stage searches for the constrained parameters and the inner stage generates the batch for the unconstrained parameters. The authors proof the average regret of the nested algorithm vanishes superlinearly. The proposed algorithms are evaluated on both benchmark functions and real problems. The process-constrained BO is a practical problem that people may face for real world problems. The authors propose an intuitive algorithm that can nicely address this BO problem, which produces good performance for both benchmark functions and real world problems.

Reviewer 2



The authors propose a new method for Bayesian optimization that allows to fix some of the variables of the input domain before computing a batch. This is practical in real experiments and the authors mentions some real cases like the optimization of heat treatments in the metallurgy industry or nano fibre production. The paper is very well motivated and written, the hypothesis and challenges of the problem are clear and the applicability of the proposed methodology in real-world scenarios is evident. In the theoretical analysis I didn't see any major flag. The experimental section is not very extensive but this is reasonable in this case as this is the first method that allow to freeze some variables before constructing a batch. I think that this paper can be a good contribution to NIPS as it correctly formalizes a solution to a practical problem that hasn't been addressed before. I enjoyed reading this work. My only recommendation for the authors to improve the impact of their work is the publication of the code needed to run their method, as this would increase the impact and visibility of this paper. Also, I would like to see if the authors think that this method can be extended to other acquisitions and if they have plans to do so.

Reviewer 3



The paper presents a bayesian optimization framework in which samples can be collected in similarly constrained batches. In short, rather than selecting a single point to sample a label from (as in usual bayesian optimization) or selecting multiple points simultaneously (as in batch bayesian optimization), multiple samples in a batch can be collected in which one or more variables need to remain constant throughout the batch. This is motivated from experiments in which some parameter (e.g., temperature in an oven, geometry of an instrument) can be designed prior to sampling, and be reused in multiple experiments in a batch, but cannot be altered modified during a batch experiment. The authors (a) propose this problem for the first time (b) propose two algorithms for solving it, (b) provide bounds on the regret for the most sophisticated of the two algorithms, and (c) present a comparison of the two algorithms on a variety of datasets. One experiment actually involves a metallurgical setup, in which the authors collaborated with researchers in their institution to design a metal alloy, an interdisciplinary effort that is commendable and should be encouraged. My only concern is that the paper is somewhat dense. It is unlikely that someone unfamiliar with Bayesian optimization would be able to parse the paper. Even knowing about bayesian optimization, notation is not intoduced clearly and the algorithms are hard to follow. In particular: -what is sigma in Algorithms 1, and 2. Is it the variance of the GP used in the UCB? -The activation fuction alpha^GB-UCB should be clearly and explicitly defined somewhere. -The description of the inner loop and the outer loop in Algorithm 1 are not consistent with the exposition in the text: h and g are nowhere to be found in Algorithm 1. This seems to be because your replace the optimization with the GP-UCB-PE. It would be better if the link is made explicit. -There is absolutely no description of how any optimization involving an activation function is performed. Do you use sampling? Gradient descent? Some combination of both? These should be described, especially in how they where instantiated in the experiments. -Is alpha different from alpha^GP-UCB in Alg. 1? how? -In both algorithms 1 and 2, it is not clear how previously selected samples in a batch (k'\< k) are used to select the k-th sample in a batch, other than that they share a constrained variable. All of these are central to understanding the algorithm. This is a pity, as the exposition is otherwise quite lucid.